# Nutritional and Phytochemical Characterization of Freeze-Dried Raspberry (*Rubus idaeus*): A Comprehensive Analysis

**DOI:** 10.3390/foods13071051

**Published:** 2024-03-29

**Authors:** Mirko Marino, Claudio Gardana, Marco Rendine, Dorothy Klimis-Zacas, Patrizia Riso, Marisa Porrini, Cristian Del Bo’

**Affiliations:** 1Department of Food, Environmental and Nutritional Sciences (DeFENS), Università degli Studi di Milano, Via Celoria 2, 20133 Milano, Italy; claudio.gardana@unimi.it (C.G.); marco.rendine@unimi.it (M.R.); patrizia.riso@unimi.it (P.R.); marisa.porrini@unimi.it (M.P.); cristian.delbo@unimi.it (C.D.B.); 2School of Food and Agriculture, University of Maine, 232 Hitchner Hall, Orono, ME 04469, USA; dorothea@maine.edu

**Keywords:** berry fruits, nutrients, phytochemicals, (poly)phenols, UHPLC-DAD-HR-MS, chemical profiling

## Abstract

Several studies have highlighted the beneficial effects of consuming red raspberries on human health thanks to their high content of phytochemicals. However, the products used in these studies, both in the raw or freeze-dried form, were not fully characterized for nutrient and phytochemical composition. In this study, we aimed to determine the nutrient and non-nutrient compounds present in a freeze-dried red raspberry powder widely used by the food industry and consumers. The main sugars identified were fructose (12%), glucose (11%), and sucrose (11%). Twelve fatty acids were detected, with linoleic acid (46%), α-linolenic acid (20%), and oleic acid (15%) being the most abundant. Regarding micronutrients, vitamin C was the main hydro-soluble vitamin, while minerals, potassium, phosphorous, copper and magnesium were the most abundant, with concentrations ranging from 9 up to 96 mg/100 g, followed by manganese, iron and zinc, detected in the range 0.1–0.9 mg/100 g. Phytochemical analysis using UHPLC-DAD-HR-MS detection revealed the presence of Sanguiin H6 (0.4%), Lambertianin C (0.05%), and Sanguiin H-10 isomers (0.9%) as the main compounds. Among anthocyanins, the most representative compounds were cyanidin-3-sophoroside, cyanidin-3-glucoside and cyanidin-3-sambubioside. Our findings can serve as a reliable resource for the food industry, nutraceutical applications and for future investigations in the context of human health.

## 1. Introduction

Consumption of berry fruits, such as raspberries, has been reported to confer protection against hyperglycemia, dyslipidemia [1] hypertension [2], liver function [3], vascular function and cardiovascular disease [4], and certain types of cancer [5]. These beneficial effects are often attributed to the presence of non-nutritional components, particularly (poly)phenols [6]. Recently, Popovic et al. [7] emphasized that the inclusion of raspberry pomace, rich in (poly)phenols, unsaturated fatty acids, and fiber in the regular diet, positively reduced certain cardiovascular risk factors and liver function indicators. Therefore, studies evaluating the beneficial properties of raspberry fruit, or its extracts should consider both their nutrient and non-nutrient content. 

It has been reported that red raspberries contain nutrients such as sugars, fats, proteins, and vitamins, as well as non-nutrients such as fiber, organic acids, and (poly)phenols, particularly anthocyanins and ellagitannins [8]. Anthocyanins contribute to the characteristic red-purple color of raspberries, with cyanidin-3-O-glucoside, cyanidin-3-O-sophoroside, and cyanidin-3-O-sambubioside being the primary anthocyanins depending on the cultivar [9]. Ellagitannins, on the other hand, are characterized by the presence of one or more hexahydroxydiphenoyl (HHDP) moieties esterified with glucose. The diverse modes of linkage between HHDP residues and the glucose moiety result in a highly variable molecular structure [10]. The major ellagitannins in raspberries are sanguiin H-6 and lambertianin, which are dimers and trimers of casuarictin, respectively. These compounds are formed through intermolecular C-O bonds between an HHDP group and a galloyl residue. The content of sanguiin H-6 and lambertianin in whole raspberry fruit ranges from 360–750 mg/kg and 280–630 mg/kg, respectively [11]. Despite the recognized health benefits of red raspberries, comprehensive research examining their nutritional and non-nutritional constituents remains limited, especially for the freeze-dried product. In fact, the available data were derived from studies analyzing the nutritional content of raw products and/or derivates specifically for the content of macronutrients (e.g., total sugars, total proteins, and total lipids), while only partial information is available for single macronutrients. To the best of our knowledge, limited data are also accessible for minerals and vitamins, including also bioactive compounds. In addition, the latter are commonly quantified by indirect methodologies as total compounds (e.g., total polyphenols, total anthocyanins), and there is little information related to the fate of the single compounds contained in the raspberries. Therefore, the aim of this study was to provide a comprehensive characterization of freeze-dried raspberry (*Rubus idaeus*), considering both its nutritional and non-nutritional components. While the health benefits of red raspberries have been widely recognized, there remains a gap in understanding the full spectrum of their nutritional and non-nutritional constituents. This characterization can serve as a valuable resource for future in vivo and in vitro studies, enabling a deeper understanding of the potential health benefits and facilitating the development of innovative applications in the field of nutraceutical research.

## 2. Materials and Methods

### 2.1. Chemicals

Standards of cyanidin (Cy)-, pelargonidin (Pel)- and their 3-O-glucoside (glc), Cy-rutinoside (Cy-rut), Cy-3,5-di-glucoside (Cy-di-glc), Cy-sambubioside (Cy-sam), and Cy-3-O-sophoroside (Cy-sop) were purchased from Polyphenols Laboratory (Sandnes, Norway). Potassium chloride, hydrochloric acid, methanol, acetonitrile, acetone, Folin reagent, phosphoric and trifluoroacetic acid (TFA) were from Merck (Darmstadt, Germany). Ascorbic acid, 4-dimethylamino-cinnamaldehyde (DMAC), ammonium acetate, acetic acid, glucose, fructose, saccharose, citric, isocitric, succinic, malic, fumaric, oxalic, tartaric, cis-aconitic and quinic acid were provided by Sigma-Aldrich (St. Louis, MO, USA). Extrasynthese (Genay, France) supplied catechin (CAT), epicatechin (EC), procyanidin C1 (PC1), procyanidin A2 (PA2) and punicalagin. Water was obtained from the Arium pro apparatus (Sartorius, Milan, Italy). The freeze-dried raspberries (*Rubus idaeus*) were a gift from FutureCeuticals (Momence, IL, USA). 

### 2.2. Moisture, Ash and Protein Determination

Moisture content was determined by a moisture analyzer Radwag mod. MA 50.R (Vetrotecnica, VR, Italy). Ash determination was carried out according to the AOAC method (AOAC, 2005).

The nitrogen (N) content was determined by conventional acid hydrolysis and Kjeldahl digestion, using a copper catalyst in 2 g of raspberry powder (RP). The ammonia was distilled and collected in a solution of boric acid, which was then titrated against standard acid. Digestion and distillation were carried out using a Kjeltec 1002 apparatus (Foss, MI, Italy). The protein content was calculated as total N × 5.6.

### 2.3. Mineral Determination

Approximately 400 mg of RP was mineralized at 120 °C in 5 mL 14.4 M HNO_3_, clarified with 1.5 mL H_2_O_2_ 33% (*w*:*v*) and finally dried at 80 °C. The mineralized material was dissolved in 5 mL of a solution 0.1 M HNO_3_ and filtered on a 0.45 µm nylon membrane. Metal content was determined by ICP-MS model 7850 (Agilent, Milan, Italy).

### 2.4. Organic Acids Determination

Citric, isocitric, succinic, malic, fumaric, oxalic, tartaric, cis-aconitic, α-ketoglutaric and quinic acid were measured by the UHPLC-UV method according to Baccichet et al. [12]. Briefly, 0.5 g of the RP was weighed in a 10 mL tube, 5 mL of a solution 0.5% EDTA in water was added and the mixture was sonicated for 10 min. The mixture was centrifuged at 1650× *g* for 10 min and the supernatant recovered. The residue was extracted by 4 mL of a solution 0.5% EDTA in water and treated as described above. The supernatants were combined, and then the final volume was adjusted to 10 mL with water. Finally, 5 µL was injected into the chromatographic system. The OA analysis was performed using a Vanquish Flex UHPLC separation module (Thermo Scientific, MA, USA) coupled with a model Q-Exactive Orbitrap HR-MS equipped with a HESI-II probe (Thermo Scientific) set in negative ion mode. A 1.8 μm HSS T3 column (150 × 2.1 mm, Waters) was used for separation at a flowrate of 0.2 mL/min. The eluents were 0.02% HCOOH in water (A) and CH_3_CN (B). The following elution gradient was used: 0% B for 7 min, 0–80% B in 1 min, 80% B for 3 min. Then, we returned it to initial conditions in 1 min. The column and samples were maintained at 30 and 20 °C, respectively. A total of 1 μL was injected in the UHPLC system. The MS conditions were as follows: the spray voltage was −3.0 kV, capillary was −32 V, tube lens was −80 V, sheath gas flow rate was 55 (arbitrary units) and auxiliary gas flow rate was 15 (arbitrary units). The temperature for the capillary and the heater was 320 and 120 °C, respectively. The analysis was performed in full scan mode in the range (*m*/*z*)^−^ 50–500 u. The resolution, AGC target, maximum ion injection time and mass tolerance were 70 K, 1× 10^6^, 100 ms and 2 ppm, respectively. The ion with *m*/*z* 91.0038 u, corresponding to the formic acid dimer [2M − H]^−^, was used as the lock mass. The MS data were processed using Xcalibur™ 4.3 software release (Thermo Scientific). Calibration curves were performed in the range 1–20 µg/mL. The organic acid content in the sample was expressed as g/100 g RP.

### 2.5. Lipid Determination

#### 2.5.1. Total Lipid

Soxhlet extraction was achieved with 6 g of RP powder and 10 g of Na_2_SO_4_ using a Soxtec HT 1043 system (Foss, Milan, Italy) containing 180 mL of a solution of ethyl ether/petroleum ether (1:1, *v*/*v*). The mixture was extracted at 140 °C for 12 h, followed by a 30 min solvent rinse and solvent evaporation. The weighted residue was considered as the lipid content (% DW). 

#### 2.5.2. Fatty Acid Determination

Approximately 0.3 g of RP were weighed in a 10 mL tube, 2 mL of a solution chloroform/methanol (3:1, *v*/*v*) were added and the mixture was vortexed for 3 min. The mixture was centrifuged at 1650× *g* for 10 min at 20 °C and the supernatant was transferred to a screw-capped Pyrex tube. The residue was extracted and treated as described above. The extracts were combined, dried under N_2_ flow, and the residue was suspended with 2 mL of a solution methanol/toluene (4:1) and 0.2 mL of acetyl chloride. The mixture was placed in an oven at 100 °C for 1 h, cooled and 5 mL of a 6% H_2_CO_3_ solution was added. The mixture was then centrifuged at 16,500× *g* for 10 min at 20 °C and the upper clear layer recovered, diluted with hexane and analyzed by GC-FID according to Gardana et al. [13]. Chromatographic separations were achieved using an Omegawax 320 capillary column (30 m × 0.32 mm i.d.; Supelco, Milan, Italy), under the following conditions: initial isotherm, 140 °C for 5 min; temperature gradient, 2 °C/min to 210 °C; final isotherm, 210 °C for 20 min. The injector temperature was 250 °C. The injection volume was 1 μL, with a 1/100 split ratio, and the FID temperature was 250 °C. The carrier and makeup gas were H_2_ and N_2_, respectively. Fatty acid retention times were obtained by injecting the Omegawax test mix as the standard. The fatty acid content was expressed as a percentage of the total fatty acids.

### 2.6. Determination of Sugars

Approximately 100 mg of RP were dispersed in 50 mL of deionized water, and the suspension was then sonicated for 10 min and centrifuged at 1650× *g* for 5 min, and the supernatant recovered. The residue was extracted twice with 20 mL of water and treated as described above. The supernatants were combined, and then the final volume was adjusted to 100 mL with water. The extracts were diluted with a solution of acetonitrile/water (70:30, *v*/*v*) and the sugar content was assessed as described by Gardana et al. [13]. The chromatographic system was an UHPLC Vanquish model Flex (Thermo Scientific) coupled to a High-Resolution MS model Q-Exactive (Thermo Scientific) operating in negative mode. A 1.7 μm BEH Amide column (150 × 2.1 mm, Waters) was used for the separation in isocratic mode at a flowrate of 0.2 mL/min. The eluents was (A) 0.02% NH_4_OH in acetonitrile and (B) 0.02% NH_4_OH in water (A:B, 70:30, *v*/*v*). The column and the sample were maintained at 30 and 20 °C, respectively. The MS conditions were the following: spray voltage −3 kV, sheath gas 35, auxiliary gas 10, capillary temperature 275 °C, heather 120 °C, capillary voltage −37.5 V and tube lens −80 V. All data were acquired by Xcalibur software (Thermo Scientific). Acquisition was carried out in scan mode in the range 100–600 u. Calibration curves were obtained from glucose, fructose and sucrose stock solutions prepared by dissolving 20 mg of standard powder in 100 mL of water. The working solutions in water/acetonitrile (30:70, *v*/*v*) were prepared in the range of 2–50 µg/mL. A total of 1 µL was injected in the UHPLC system.

### 2.7. Determination of Total (poly)Phenols and Proanthocyanidins

Total (poly)phenols in RP were evaluated following the Folin–Ciocalteu method and utilizing gallic acid as a standard [14]. Results of triplicate analyses were given as g/100 g of gallic acid equivalents (GAE). Briefly, approximately 100 mg of RP was dissolved into 10 mL of a methanol/water (20:80, *v*/*v*) solution. The mixture was centrifuged at 1650× *g* for 10 min, and the supernatant was transferred to a 20 mL volumetric flask. The residue was extracted by 5 mL of a methanol/water (20:80, *v*/*v*) solution, and the mixture was treated as described above. The volume was adjusted to 20 mL by water. The solution was diluted by water and centrifuged at 1650× *g* for 2 min, and the analysis performed according to Gardana et al. [15]. The gallic acid calibration curve was in the range 5–100 µg/mL and the results of triplicate analyses are given as g gallic acid equivalents (GAE)/100 g RP.

Total proanthocyanidins were determined according to Gardana et al. [15]. Briefly, approximately 50 mg of the RP were dissolved in 5 mL of a solution of acetone/water/acetic acid (75:24.5:0.5 *v*/*v*/*v*). The mixture was vortexed for 30 s, sonicated for 10 min, and centrifuged at 1650× *g* for 5 min at 20 °C, and the supernatant was recovered. The residue was extracted with 4 mL of a solution of acetone/water/acetic acid (75:24.5:0.5 *v*/*v*/*v*) and treated as described above. The volume was adjusted to 10 mL with water and diluted for DMAC assays. The Procyanidin A2 (PA2) calibration curve was in the range 2–50 µg/mL and the results of triplicate analyses are given as g PA2 equivalents/100 g RP.

### 2.8. Proanthocyanidin Determination by UHPLC-DAD-Orbitrap MS

Approximately 20 mg of RP was dissolved into 5 mL of a methanol/water (20:80, *v*/*v*) solution. The mixture was centrifuged at 1650× *g* for 10 min, and the supernatant was transferred to a 10 mL volumetric flask. The residue was extracted by 4 mL of a methanol/water (20:80, *v*/*v*) solution, and the mixture was treated as described above. The volume was adjusted to 10 mL by water and the solution was centrifuged at 1650× *g* for 2 min. The analysis was performed according to Gardana et al. [15].

The analysis was performed on an Vanquish Flex UHPLC system (Thermo Scientific, Rodano, Italy) coupled with a DAD (Thermo) and a high-resolution Fourier-Transform Orbitrap mass spectrometer, Q-Exactive model Focus (Thermo), equipped with a HESI-II probe for ESI. The operative conditions were as follows: spray voltage −3.0 kV, sheath gas flow rate 55 (arbitrary units), auxiliary gas flow rate 20 (arbitrary units), capillary temperature 350 °C, capillary voltage −60 V, tube lens −100 V, and heater temperature 130 °C. A 1.7 μm BEH Shield C_18_ column (150 × 2.1 mm, Waters) maintained at 50 °C was used for the separation. The flow rate was 0.45 mL/min, and the eluents were 0.05% formic acid in water (A) and acetonitrile (B). The UHPLC separation was achieved by the following linear elution gradient: 5–35% of B for 10 min, which was then increased from 35–80% B for 10 min. The acquisition was made in the full-scan mode in the range (*m*/*z*)^−^ 100–1500 and 1000–3000 u, using an isolation window of 2 ppm. The AGC target, injection time, mass resolution, energy, and gas in the collision cell were 1 × 10^6^, 100 ms, 70 K, 30–60 V, and N_2_, respectively. The MS data were processed using Xcalibur software (Thermo Scientific). The peak identity was ascertained by evaluating the accurate mass, the fragments obtained in the collision cell, and the on-line UV spectra (200–450 nm) [15]. 

### 2.9. Extraction and Analysis of ACNs from Raspberry Powder

Anthocyanin extraction was performed as follows: 100 mg of RP was dissolved in 5 mL of a solution of methanol/2% TFA (20:80, *v*/*v*) and sonicated for 10 min. The suspension was centrifuged at 1650× *g* for 10 min, the supernatant was recovered, and the residue was extracted with the solution of methanol/2% TFA (20:80, *v*/*v*) until the color disappeared (x3). Finally, the volume was adjusted to 25 mL with aqueous 2% TFA and the solution was stored at −20 °C. The ACN total content was determined spectrophotometrically, as described by Lee et al. [16]. Anthocyanin identification was performed using an Acquity UHPLC system (Waters, Milford, MA, USA) equipped with a DAD model E-Lambda (Waters) and an HR-MS model Exactive (Thermo Scientific, Rodano, Italy) equipped with a HESI-II probe for ESI and a collision cell (HCD). A 2.6 μm Kinetex C_18_ column (150 × 4.6 mm, Phenomenex, Torrence, CA, USA) protected with guard column, carried out the separation at 1.7 mL/min, and flow-rate split 5:1 before electrospray ionization (ESI) source. The column and sample were maintained at 45 and 20 °C, respectively. The eluents were (A) 0.2% TFA in water and (B) acetonitrile/0.2% TFA in water (35:65, *v*/*v*). The linear gradient was as follows: 0–15 min 14% B; 15–25 min from 14 to 20% B; 25–35 min from 20 to 32% B; 35–45 min from 32 to 50% B; 45–48 min 50 to 90% B; and 90% for 3 min. The MS operative conditions were as follows: spray voltage +4.0 kV, sheath gas flow rate 60 (arbitrary units), auxiliary gas flow rate 20 (arbitrary units), capillary temperature 350 °C, capillary voltage +30 V, tube lens +80 V, skimmer +25 V, and heater temperature 130 °C. The acquisition was assessed in the full-scan mode in the range (*m*/*z*)^+^ 200–2000 u, using an isolation window of 2 ppm. The AGC target, injection time, mass resolution, energy, and gas in the collision cell were 1 × 106, 100 ms, 50 K, 20 V, and N_2_, respectively. The MS data were processed using Xcalibur Software (Thermo Scientific). Peaks were identified by evaluating the accurate mass, the fragments obtained in the collision cell, and the on-line UV spectra (220–700 nm). Working solutions (n = 5) were prepared in the range of 2–50 µg/mL, and 20 µL was injected into the chromatographic system. Chromatographic data were integrated at 520 nm, and each analysis was carried out in triplicate (three technical replicates) [17].

### 2.10. Extraction and Analysis of Ellagitannins from Raspberry Powder

Approximately 1 g of RP was dissolved in 10 mL of a solution methanol/water (20:80, *v*/*v*) and sonicated for 10 min. The suspension was centrifuged at 1650× *g* for 10 min, the supernatant recovered and the residue was extracted with 10 mL of solution methanol/water (20:80, *v*/*v*). The suspension was centrifuged at 1650× *g* for 10 min, the supernatant recovered, and the volume was adjusted to 25 mL with water and the solution was stored at −20 °C.

The chromatographic analysis of the ellagitannin was performed by an UHPLC Vanquish Flex (Thermo) coupled to a Vanquish HL PDA (Thermo) and an HR-MS Orbitrap mod. Exactive (Thermo) equipped with a HESI-II probe for ESI and a collision cell (HCD). The separation was carried out with a 1.7 μm BEH C18 column (150 × 2.1 mm, Waters) protected with a guard column, at 1.7 mL/min, and flow rate was 0.45 mL/min. The injection volume was 7.5 μL. The column and sample were maintained at 45 °C and 20 °C, respectively. The eluents were (A) 0.05% HCOOH in water and (B) 0.05% HCOOH in CH3CN. The linear gradient was as below: 0–20 min from 2 to 30% B, 20–30 min from 30 to 80% B, and 80% for 2 min. The MS operative conditions were as follows: spray voltage −3.0 kV, sheath gas flow rate 55 (arbitrary units), auxiliary gas flow rate 20 (arbitrary units), capillary temperature 300 °C, capillary voltage −95 V, tube lens −190 V, skimmer −46 V, and heater temperature 120 °C. The acquisition was assessed in the full-scan mode in the range (*m*/*z*)^−^ 150–3000 u, using an isolation window of ±2 ppm. The AGC target, injection time, mass resolution, energy, and gas in the collision cell were 1 × 106, 100 ms, 50 K, 80 V, and N2, respectively. The MS data were processed using Xcalibur Software (Thermo). The mass spectrometer setup was performed by infusion of a 5 μg/mL punicalagin solution. The punicalagin calibration curve in the range of 0.2–20 μg/mL was used for the ellagitannin quantification and data were corrected by the molecular weight ratio. Peaks were identified by evaluating the accurate mass, the fragments obtained in the collision cell, and the online UV spectra (220–450 nm).

### 2.11. Ascorbic Acid Determination

The ascorbic acid analysis was performed using an Alliance model 2695 (Waters, Milford, MA, USA) coupled with a DAD model 2998 (Waters). A 5 μm Atlantis T3 column (250 × 4.6 mm, Waters) was used for separation at a flowrate of 1.2 mL/min. The eluents were 1% HCOOH in water (A) and CH_3_CN (B). The following elution gradient was used: 0% B for 5 min, 0–90% B in 1 min, 90% B for 4 min. Then, it was returned to initial conditions in 1 min. The column and samples were kept at 30 and 15 °C, respectively. A total of 20 μL was injected in the HPLC system. Chromatograms were acquired in the range 220–450 nm and integrated at 246 nm. Calibration curves was performed in the range 2–50 μg/mL.

## 3. Results and Discussion

### 3.1. Nutrient Composition

The nutritional composition of the RP is reported in Table 1. Sugars were the most abundant nutrient, constituting more than 35% of the RP. The main sugars were fructose, glucose, and sucrose. They were present in comparable quantities, approximately 12% (Table 1), while rhamnose and xylose were detected in quantities below the limit of quantification. Our data were in line with those of the literature. For example, Akšić et al. [18] reported that glucose and fructose were the most abundant sugars of raspberry. The glucose content ranged from 19 to 33 g/100 g, while the fructose content ranged from 13 to 24 g/100 g. Yu et al. [19] analyzed various raspberry varieties and found that fructose and glucose were the predominant monosaccharides, while sucrose was the major disaccharide. In particular, the fructose, glucose and sucrose content were in the range of 14–37, 11–31 and 0.2 to 38 g/100 g DW, respectively. These studies provide consistent information about the presence of glucose and fructose as the most abundant sugars in raspberries; however, their content varied depending on raspberry varieties, cultivation conditions, and analytical methods. The sugar content in raspberry contributes to the taste, sweetness, and overall flavor profile of the product. In addition, the balance between sweetness and acidity represents an important factor that may influence consumer preferences. From a nutritional point of view, the presence of fructose as a fruit sugar in the RP may contribute to the regulation of the glycemic curve due to its lower glycemic index [20]. 

Raspberries contain small seeds that are normally consumed together with the fruit, thus contributing to the protein and lipid content of the product. Regarding proteins, the content was 8.1 g/100 g (Table 1); this result matched the composition reported for raw raspberries in previous studies by Burton-Freeman et al. [6] and VandenAkker et al. [3], as well as the USDA database, when accounting for the variation in water content. In addition, it seems comparable with data reported by Bushman et al. (2004), which found a protein content that ranged from 6 to 7%. Conversely, other studies have reported a protein content in the red raspberry seeds above 12% [21].

With regard to the total lipid (TLs) content, the relative percentage of SFAs, MUFAs, PUFAs, n–3, n–6, and the UFA/SFA ratio of the TLs from the RP are reported in Table 1, while the fatty acid compositions are reported in Table 2. Overall, we found that the TL content in the freeze-dried RP was 6.2% (based on the DW); in addition, we detected the presence of palmitic, oleic, linoleic and α-linolenic acid. Specifically, linoleic acid constituted about 46% of total FA, while α-linolenic and oleic acid were at about 20% and 15%, respectively. Palmitic, stearic and arachidic acids were the main SFA and the sum reached up to 10% of TL. Our findings are in line with those reported by Burton-Freeman et al. [6] and the USDA database. Other studies evaluated the lipid content of the raspberries. For example, Bushman et al. [22] reported that the red raspberry (*Rubus idaeus* L.) seed contained at about 23% fats, consisting of 54% linoleic acid (C18:2n6), 30% α-linolenic acid (C18:3n3), 11% oleic acid (C18:1n9), and 3–4% saturated fatty acids (SFA) with the predominant concentration of palmitic acid (C16:0). Bushman et al. [22] and Celik et al. [23] investigated the lipid and fatty acid composition of 11 wild-grown red raspberry genotypes and one cultivated raspberry (Heritage cultivar). The TL content ranged from 0.36 to 0.6%, with the cultivated raspberry having a higher lipid ratio than the wild genotypes. VandenAkken et al. [3] reported 0.36% of TL in whole red raspberry (*Rubus idaeus*). The main fatty acids found in all genotypes were linoleic acid (42 to 53%) and linolenic acid (18 to 24%). Wild genotypes had higher amounts of linoleic, palmitic, and stearic acid compared to the cultivated raspberry, while the cultivated raspberry had a higher oleic acid content [23]. PUFAs were the predominant fatty acids, representing between 67 and 76% of TL content. Vara et al. [24] reported a total of 21 fatty acids in the red raspberry “Kweli” cultivar. MUFA and PUFA were detected in roughly equivalent proportions by constituting approximately 58% of the TL fraction in red raspberries. The main FAs were oleic (C18:1n–9; 27%), α-linolenic (C18:3n–3; 17%) and linoleic acid (C18:2n–6; 12%). In addition, the analysis revealed a substantial presence of SFAs (at about 42%), primarily palmitic (C16:0), stearic (C18:0), arachidic (C20:0), and behenic (C22:0) acids. Both studies also highlight the prevalence of unsaturated fatty acids in the raspberry’s lipid fraction, with an UFA/SFA ratio of 7.7, which is considered an elevated amount and represents an indicator of fat nutritional quality of the diet [25].

Raspberries are also recognized to be an important food source of vitamin C. It is recognized that 100 g of berries provide almost 50% of the recommended daily allowance. The amount of ascorbic acid detected in the raspberry powder was about 112 mg/100 g (Table 1). These data were in accordance with Skrovankova et al. [26] who reported a content of ascorbic acid in the range of 5–40 mg/100 g FW, corresponding approximately to 25–200 mg/100 g DW, and with Rao et al. [27], who reported a content of vitamin C of 26.2 mg/100 g FW. 

Raspberries are also a good source of minerals and in particular Mg, K, Cu, and Fe [28]. In addition, some studies suggest raspberries as an important food source of Mn [27]. In the present study, we found that the mineral content of the freeze-dried *Rubus idaeus* was approximately 126 mg/100 g. K, P, Cu and Mg were the most abundant compounds, respectively, (range 9.1–95.6 mg/100 g), followed by Mn, Fe, Al and Zn detected in the range 0.1–0.9 mg/100 g (Table 3). The other minerals were present in amounts less than 0.1 mg/100 g, while Se was detected in trace amounts (on average 0.1 µg/100 g). Overall, our data seem in line with those of the USDA database and the literature by confirming raspberry as a berry with the major content of K [27,29] and Mn [27].

### 3.2. Non-Nutrient Composition

A large variation in the total (poly)phenol content (TPC) has been reported for red raspberry cultivars also depending on the different geographical areas. For example, TPC in Lithuanian red raspberries ranged from 279 to 504 mg/100 g for the cultivars Pokusa and Glen Moy, respectively [30]; in Finland, TPC ranged from 192 mg/100 g for the cultivar Gatineu and up to 359 mg/100 g for the cultivar Ville [31]. Finally, in Hungary the TPC content of red raspberry varied from 160 up to 336 mg/100 g [32]. In the present study, we found that TPC, evaluated by the Folin method, was 3.3 g GAE/100 g in the freeze-dried raspberry, corresponding to about 600 mg GAE/100 g FW. This value is comparable to what was found by Isik et al. [33] for the Heritage cultivar (750 mg GAE/100 g FW) and by Weber et al. [34] for the Encore cultivar (645 mg/100 g), but considerably higher than that reported by Jiang et al. [35] and by VandenAkker et al. [4], who found 73 mg GAE/100 g FW in the cultivar Chilcotin and 1.5 g GAE/100 g, respectively. Besides the fact that TPC may be influenced by the genotype and the geographical area, another important factor seems to be the period of harvesting. In fact, some studies report that late-harvest raspberries contain significantly higher TPC. In particular, higher TPC was found in the second harvest of the autumn-bearing red raspberry cultivars Heritage and Autumn Bliss; thus, it cannot be excluded that this wide variability in the TPC content could be attributed to the harvesting seasons [36]. In addition, it is important to underline that TPC in the raspberry may be affected (both under and/or overestimated) by the presence of other bioactives, including, for example, vitamin C. 

In general, the major phenolic compounds of raspberries are proanthocyanidins (PACs), ellagitannins and anthocyanins. The total amount of PACs, as determined by the DMAC assay, was on average 0.7 g/100 g RP. This amount was comparable (0.7 vs. 0.9%) to that found by Shi et al. [37], especially considering that data were obtained with different methods.

Regarding anthocyanins, the major constituents of red raspberries are cyanidin-3-sophoroside, cyanidin-3-glucoside and cyanidin-3-sambubioside. Figure 1a shows the chromatogram, extracted at 520 nm, related to the analyzed raspberry powder, while in Table 4 its individual anthocyanin content is reported. Anthocyanins were identified by co-chromatography, online UV-Vis spectra, accurate mass and fragment ions obtained by collision-induced dissociation. Six peaks (Figure 1a) with absorption maxima at 520 nm and main fragment ions at *m*/*z* 287.0552, corresponding to cyanidin (Cy), were found in the freeze-dried raspberry. Peak 1, *m*/*z* 611.1609 u, gave fragment ions with *m*/*z* 449.1080 and 287.0552 u, corresponding to the loss of two hexoside residues. Thus, considering the results obtained and what was reported in the literature [38], peak 1 was tentatively identified as Cy-3,5-O-diglucoside and its identity afterward was confirmed by the reference standard. Peak 2, *m*/*z* 611.1609 u, gave fragment ions with *m*/*z* 287.0552 u, corresponding to the loss of the sophorosyl residue. Thus, peak 2 was Cy-3-O-sophoroside and its identity then confirmed by the reference standard. Likewise, peaks 3, 4 and 5 were identified as Cy-3-O-glucoside, Cy-3-sambubioside and Cy-3-O-rutinoside, respectively. The total ACN content was 530 ± 20 mg/100 g and cyanidin-3-O-sophoroside and cyanidin-3-O-glucoside were the main components, constituting approximately 77 and 18% of total ACNs, respectively. In addition, significantly lower amounts of cyanidin-sambubioside, cyanidin-rutinoside and cyanin were detected, respectively, while the aglycones were not detected. The average contents of these ACNs were comparable to those reported by Wu et al. [39] and VandenAkker et al. [3], but higher than those reported by Ludwig et al. [40]. It should be noted that contrary to what was reported by these authors, we did not detect the presence of pelargonidin-based ACNs. However, this result seems in line with the findings showing that pelargonidin glycoside is found only in smaller quantities by constituting less than 2% of the total ACN content in the raw fruit [27]. Thus, we cannot exclude that the freeze-dried process could have negatively affected pelargonidin concentration. 

Several (poly)phenols were detected in the freeze-dried raspberry powder (Table 5) and the main ones were sanguiin H-6 (SH-6), lambertianin C (L-C) and sanguiin H-10 isomers (SH-10), whose quantities were on average 450, 60, and 94 mg/100 g, respectively. The detected amount of these compounds is comparable to what was reported by Gasperotti et al. [11], considering the amount of water present in the raw fruit, about 85%. Concerning SH-6, L-C, SH-10 and the other ellagitannins, the high-resolution mass spectrometer is essential as they mainly occur as doubly charged ions. Thus, in Figure 1b the peak with RT 12.4 min has in its isotopic distribution the ions with *m*/*z* 1870.1471 and 933.5680 u and the latter suggests that they are bi-charged ions. After fragmentation at different collision energies ions with *m*/*z* 933.0635, 633.0723 and 300.9988 u were detected, corresponding to losses of M-(galloyl-diHHDP-glucose), 933-HHDP and 633-galloylglucose, respectively. Overall, the mass data suggest that peak 13 corresponds to SH-6. Similarly, peak 12 showed mainly a double-charged ion with *m*/*z* 1401.6062 u. After fragmentation at higher collision energy (80 eV) ions with *m*/*z* 633.0723 and 300.9988 u (ellagic acid) were detected, suggesting the compound was LC. Peaks 5, 10 and 11 had a main ion with *m*/*z* 1567.1400 u and its double-charged ion 733.0683 u. All Ion Fragmentation (AIF) generated ions with *m*/*z* 933.0630, 783.0683, 633.0726 and 300.9989 u suggesting those peaks were SH-10 isomers. Peaks 2 and 3 had *m*/*z* 783.0690 which after fragmentation generated the ion with *m*/*z* 300.9989 u corresponding to ellagic acid suggesting that they were pedunculagin isomers. Peak 14 corresponded to ellagic acid conjugated to a pentoside and its amount was about 0.05%, while ellagic and gallic acid were not detected. In general, free ellagic acid levels are generally low in Rubus fruits constituting only 1–2.2% of the total ellagic acid content [30]. On the contrary, Wada et al. [41], reported high levels of free ellagic acid (40–50%) of the total content; however, the total amount of ellagic acid was about 0.04%, a value comparable to that reported in the literature. Overall, ellagitannin identification would not have been possible with unitary resolution spectrometers like quadrupoles and ion traps.

Raspberries also contain a wide range of organic acids that contribute to both the taste and acidity of the product. The total organic acids (OAs) content of the freeze-dried powder was 5.8 ± 0.2 g/100 g. Citric acid (4.9 ± 0.2 g/100 g) was the most abundant OA in freeze-dried raspberry powder, accounting for about 85% of the total OAs. Succinic, tartaric and malic acid were detected in low amounts, representing on average 6.7, 4.9 and 3.6% of the total OAs, respectively. Fumaric, oxalic and α-ketoglutaric acids represented less than 0.005%. These findings agree with those reported by Çekiç et al. [42]. Conversely, Yu et al. [19] found amounts of citric acid in the range of 3–14 g/100 g DW after the analysis of 24 raspberry cultivars obtained from various regions and countries. This variability in the OA content could be due to the variety of raspberries as well the inverse proportionality to the ripening stage.

## 4. Conclusions

In conclusion, freeze-dried raspberry represents an excellent source of nutrients and bioactive compounds, specifically vitamin C and minerals (like potassium and magnesium), whose contents seem comparable to those of raw raspberries provided by the main databases. Furthermore, freeze-dried raspberry represents a valuable source of polyphenols like proanthocyanidins ellagitannins, anthocyanins and phenolic acids. In particular, among ellagitannins, sanguiin H-6, lambertianin C and sanguiin H-10 isomers are the main compounds, while among anthocyanins, the most dominant compounds are represented by cyanidin-3-sophoroside, cyanidin-3-glucoside and cyanidin-3-sambubioside. The nutritional information is very important for the food industry, which uses freeze-dried products in the formulation of many foods (i.e., jams, jellies, purées, syrups, juices, and several bakery and dairy products) and in food labeling. In addition, the information provided may be also useful for the formulation of new ingredients and food products with high added value, as well as for the design of functional foods and nutraceuticals for population groups with specific nutritional needs. Overall, these results fill a significant gap in current research concerning the identification and quantification of raspberry constituents. Additionally, they open avenues for further investigations, including future perspectives of studying the stability of these compounds during processing and storage, which could provide valuable information for optimizing preservation techniques and improving shelf life of raspberry-derived products. Finally, it may be useful for scientists studying the potential applications of red raspberries and their use in animal and human dietary intervention studies to explore their potential health benefits and mode of action. 

## Figures and Tables

**Figure 1 foods-13-01051-f001:**
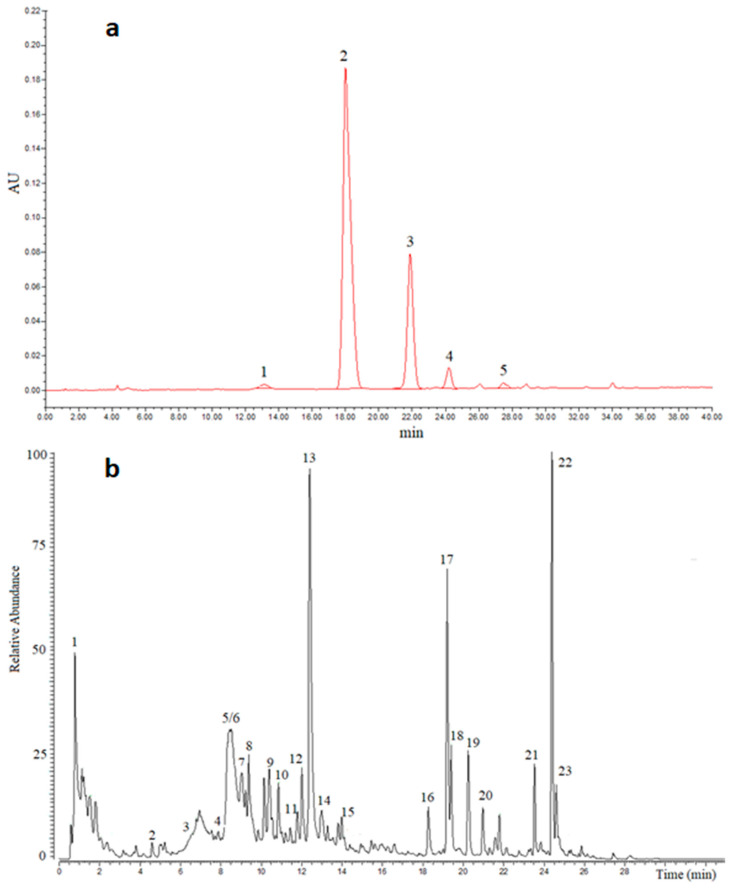
(**a**) Chromatographic profile of the ACNs in a raspberry freeze-dried powder (*Rubus idaeus*) detected at 520 nm. 1: Cyanidin-di-glucoside, 2: Cyanidin-sophoroside, 3: Cy glucoside, 4: Cy-sambubioside, 5: Cyanidin-rutinoside. See Table 4 for peak identification. (**b**) Total Ion Chromatography (TIC) in the range 150–3000 u obtained for the RP-UHPLC-HR-MS analysis of freeze-dried raspberry fruit. See Table 5 for peak identification.

**Table 1 foods-13-01051-t001:** Nutritional composition of the raspberry freeze-dried powder.

Component	%
Raw berries moisture	85.1 ± 2.3
RP moisture	6.1 ± 0.3
Ash	4.1 ± 0.3
Protein	8.1 ± 0.4
Total lipids	6.2 ± 0.5
SFA	10.6 ± 0.3
MUFA	15.6 ± 0.2
PUFA	65.9 ± 1.1
*n*–3	19.8 ± 0.3
*n*–6	46.1 ± 0.4
UFA/SFA	7.7 ± 0.2
Sugars	35.4 ± 0.3
Fructose	12.3 ± 0.3
Glucose	10.8 ± 0.1
Sucrose	12.2 ± 0.3
Ascorbic acid	0.112 ± 0.001
Total minerals	0.13 ± 0.01
Organic acids	5.8 ± 0.2
Total anthocyanins	0.53 ± 0.02
Total (poly)phenols ^a^	3.3 ± 0.2
Total PACs ^b^	0.71 ± 0.05
Total ellagitannins ^c^	0.63 ± 0.12

^a^ as Gallic Acid Equivalent (g GAE/100 g RP), ^b^ as Procyanidin A2 equivalent (g PA2 equivalent/100 g RP), ^c^ quantified on the punicalagin calibration curve and the amount corrected by the molecular weight ratio (MW component/MW punicalagin): proanthocyanidins, RP: raspberry; MUFA: monounsaturated fatty acids; PUFA: polyunsaturated fatty acids; SFA: saturated fatty acids; UFA: sum of unsaturated fatty acids (MUFA + PUFA); n3: omega 3 fatty acids; n6: omega 6 fatty acids.

**Table 2 foods-13-01051-t002:** Fatty acid composition in the raspberry freeze-dried powder.

FA	%
caproic acid	0.41 ± 0.05
lauric acid	0.19 ± 0.01
palmitic acid	5.73 ± 0.02
stearic acid	1.73 ± 0.09
oleic acid	14.75 ± 0.03
vaccenic acid	0.84 ± 0.04
linoleic acid	46.14 ± 0.44
α-linolenic acid	19.80 ± 0.29
arachidic acid	1.42 ± 0.01
behenic acid	0.71 ± 0.02
tricosanoic acid	0.21 ± 001
lignoceric acid	0.24 ± 0.06

Data are reported as percentage composition (mean ± SD). FA, fatty acids.

**Table 3 foods-13-01051-t003:** Amount of minerals in raspberry freeze-dried powder.

Mineral	mg/Kg
potassium	959.6 ± 79.7
phosphorus	121.7 ± 10.9
calcium	109.0 ± 8.6
magnesium	91.7 ± 7.2
manganese	8.2 ± 0.5
iron	6.5 ± 0.5
aluminum	5.1 ± 0.8
zinc	1.5 ± 0.2
sodium	0.52 ± 0.04
copper	0.44 ± 0.04
chrome	0.041 ± 0.005
molybdenum	0.032 ± 0.009
nichel	0.27 ± 0.04
cobalt	0.019 ± 0.002
cadmium	0.012 ± 0.001
plumb	0.010 ± 0.002
arsenic	0.006 ± 0.001
selenium	ND

Data are reported as mean ± SD. ND: not detected.

**Table 4 foods-13-01051-t004:** Anthocyanins in the raspberry freeze-dried powder.

Peak	λmax	[M]^+^	Fragment Ions	Anthocyanin	mg/100 g RP
1	520	611.1609	449.1081, 287.0552	Cy-di-Glc	5.8 ± 0.2
2	520	611.1609	287.0551	Cy-Sop	406.3 ± 15.0
3	520	449.1080	287.0553	Cy-Glc	93.7 ± 3.3
4	520	581.1504	449.1081, 287.0552	Cy-Sam	18.1 ± 0.6
5	520	595.1660	449.1081, 287.0552	Cy-Ru	6.2 ± 0.2

Data are reported as mean ± SD. Cy: cyanidin; Glc: glucose; Sop: sophorose; Sam: sambubiose; Rut: rutinose.

**Table 5 foods-13-01051-t005:** Retention time, deprotonated and fragment ions of the compounds in the raspberry freeze-dried powder.

Peak	RT	[M–H]^−^	Brute Formula	Fragments	Brute Formula	Compound
1	0.9	191.0196	C_6_H_7_O_7_			Citric acid
2	4.6	783.0690	C_34_H_23_O_22_	300.9989	C_14_H_5_O_8_	Pedunculagin isomer
3	6.7	783.069	C_34_H_23_O_22_	300.9989	C_14_H_5_O_8_	Pedunculagin isomer
4	7.8	577.1352	C_30_H_25_O_12_	288.0640	C_15_H_13_O_6_	Procyanidin dimer B-type
5	8.3	783.0690 ^a^ 1567.1416	C_34_H_23_O_22_	300.9989	C_14_H_5_O_8_	Sanguiin H-10 isomer
6	8.5	627.1561 609.1461	C_27_H_31_O_17_ C_27_H_29_O_16_	491.1405 284.0323	C_20_H_27_O_14_ C_15_H_8_O_6_	Eriodictyol-di-glucoside
7	9.0	577.1352	C_30_H_25_O_12_			Procyanidin Dimer B-type
8	9.4	627.1565	C_27_H_31_O_17_	491.1405 285.0402	C_20_H_27_O_14_ C_15_H_9_O_6_	Eriodictyol-di-glucoside
9	10.4	561.1401	C_30_H_25_O_11_	289.0717	C_15_H_13_O_6_	Procyanidin dimer B-type
10	10.8	783.0694 ^a^ 1567.1416	C_34_H_23_O_22_ C_68_H_47_O_44_	300.9989	C_14_H_5_O_8_	Sanguiin H-10 isomer
11	11.8	783.0694 ^a^ 1567.1416	C_34_H_23_O_22_ C_68_H_47_O_44_	300.9989	C_14_H_5_O_8_	Sanguiin H-10 isomer
12	12.0	1401.6062 ^a^	C_123_H_79_O_78_	633.0723 469.0042 300.9988	C_27_H_19_O_18_ C_21_H_9_O_13_ C_14_H_5_O_8_	Lambertianin C
13	12.4	1870.1482 934.0715 ^a^	C_82_H_53_O_52_ C_41_H_26_O_26_	933.0635 633.0723 469.0042 300.9988	C_41_H_25_O_26_ C_27_H_19_O_18_ C_21_H_9_O_13_ C_14_H_5_O_8_	Sanguiin H-6
14	13.0	433.041	C_19_H_13_O_12_	300.9988	C_14_H_5_O_8_	Ellagic acid pentoside
15	14.0	567.2083	C_27_H_35_O_13_			Saponin
16	18.3	677.2831 1355.5713 ^b^	C_34_H_45_O_14_ C_68_H_89_O_28_	489.0673 315.0145	C_22_H_17_O_13_ C_15_H_7_O_8_	Saponin
17	19.2	679.3696 1359.7452 ^b^	C_36_H_55_O_12_ C_72_H_111_O_12_	517.3165	C_30_H_4_ O_7_	Tenuifolin
18	19.4	679.3696 1359.7452 ^b^	C_36_H_55_O_12_ C_72_H_111_O_12_	517.3165	C_30_H_45_O_7_	Tenuifolin
19	20.2	711.3967	C_37_H_59_O_13_	503.3377	C_30_H_47_O_6_	Triterpenoid glycoside (+HCOOH)
20	21.0	709.3811	C_37_H_57_O_13_	501.3220	C_30_H_45_O_6_	Triterpenoid glycoside (+HCOOH)
21	23.5	695.4012	C_37_H_59_O_12_	487.3426	C_30_H_47_O_5_	Unknown
22	24.4	679.3696 1359.7452 ^b^	C_36_H_55_O_12_ C_72_H_111_O_12_	517.3165	C_30_H_45_O_7_	Tenuifolin
23	24.6	1387.7402 679.3696	C_36_H_55_O_12_	679.3696	C_36_H_55_O_12_	Triterpenoid glycoside (+HCOOH)

^a^: doubled charged ions [M–2H]^−^, ^b^: dimers [2M–H]^−^.

## Data Availability

The original contributions presented in the study are included in the article, further inquiries can be directed to the corresponding author.

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
