# Peer review of "Nutritional and Phytochemical Characterization of Freeze-Dried Raspberry (Rubus idaeus): A Comprehensive Analysis"

_foods, 2024, doi:10.3390/foods13071051_

Round 1

Reviewer 1 Report

Comments and Suggestions for Authors

The comments for the authors:

1. More key findings should be highlighted in the Abstract section.

2. The importance and the aim of the study should be more emphasised. What is the novelty of the study?

3. "Results of triplicate analyses are given as g/100g of gallic acid equivalents (GAE)." You mean g GAE/ 100 g extract or sample? Please add the unit for every content analysed in the section Materials and methods.

4. Tables should be uniform.

5. What is the future perspective? What is the practical application of the finding. Please, make a suggestion.

6. The authors should check the text for typos and grammatical errors.

Author Response

  1. More key findings should be highlighted in the Abstract section.

Answer: We thank the reviewer; we have implemented the abstract section by providing more key findings.

  1. The importance and the aim of the study should be more emphasised. What is the novelty of the study?

Answer: We have noted these in the introduction and the conclusion sections of the manuscript by explaining the relevance of the study aim and discussing the novelty of our findings.

  1. "Results of triplicate analyses are given as g/100g of gallic acid equivalents (GAE)." You mean g GAE/ 100 g extract or sample? Please add the unit for every content analysed in the section Materials and methods.

Answer: In accordance with the reviewer's suggestion, we have modified it for better understanding.

  1. Tables should be uniform.

Answer: Done.

  1. What is the future perspective? What is the practical application of the finding. Please, make a suggestion.

Answer: We thank the reviewer; we have provided clear perspectives and practical applications of our findings in the conclusion section. Also, this aspect has been implemented in the abstract section.

  1. The authors should check the text for typos and grammatical errors.

Answer: The manuscript has been revised for typos and grammatical errors.

Reviewer 2 Report

Comments and Suggestions for Authors

1. The methods of quantitative analysis are described poorly. For example, mineral determination, organic acid assay, UHPLC, etc. It is advisable to present information about an equipment, chromatographic columns, conditions of chromatography and so on. When it is presented just a reference, the manuscript is difficult to be read.

2.       L202. There are raspberry seeds described, but previously no information about them. How were they isolated? What is their origin?

3.       The origin of raw materials should be added to the manuscript.

4.       L239. There is the data about vitamin C analysis, but no information about the method used in the paragraph material and methods.

5.       In the conclusions vitamin K is mentioned, but in the text of the manuscript nothing about it. How can it be?

6.       I discuss the novelty of the research should be highlighted.

7.       In the references the instruction part should be removed.

Author Response

  1. The methods of quantitative analysis are described poorly. For example, mineral determination, organic acid assay, UHPLC, etc. It is advisable to present information about an equipment, chromatographic columns, conditions of chromatography and so on. When it is presented just a reference, the manuscript is difficult to be read.

Answer: We thank the reviewer; we have implemented it in the materials and methods section based on the reviewer's suggestion.

  1. L202. There are raspberry seeds described, but previously no information about them. How were they isolated? What is their origin?

Answer: For raspberry seeds we refer to the seeds normally contained in each fruit and consumed together with the pulp. They were not isolated from the freeze-dried since they were ground in the food matrix. We have clarified this aspect.

  1. L239. There is the data about vitamin C analysis, but no information about the method used in the paragraph material and methods.

Answer: We thank the reviewer. We have added the paragraph as requested (2.11).

  1. In the conclusions vitamin K is mentioned, but in the text of the manuscript nothing about it. How can it be?

Answer: We apologize to the reviewer. It was not vitamin K but potassium (K). We have corrected the sentence.

5.I discuss the novelty of the research should be highlighted.

Answer: We have implemented it in the introduction and the conclusion sections of the manuscript by better discussing the novelty of our findings.

  1. In the references the instruction part should be removed.

Answer: The reviewer is right. We have removed the instructions.

Reviewer 3 Report

Comments and Suggestions for Authors

1. The expression of chemical structure formulas of chemical substances in this paper needs to be further corrected and standardized.

2. The use of units in this paper is not standardized, for example, ml and mL are mixed, and there is a lack of uniformity in the presence or absence of spaces in front of the units, etc., which needs to be further revised.

3. It is recommended that the authors reorder Tables 2 and 3 in this paper; the order of the tables should be in the order in which they appear in the text.

4. It is recommended that the authors add the names of the corresponding fatty acids in Table 3, which would be more clear and concise, rather than just providing them in the table notes.

5. The presentation of Table 5 is not standardized and the authors are advised to revise it to a 3-line table.

6. After "3. Results and Discussion" is "5. Conclusions"? It is suggested that the authors amend the expression.

Comments on the Quality of English Language

1. The standardized use of abbreviations, units, and chemical structure formulas, etc. in this paper needs to be further corrected

Author Response

  1. The expression of chemical structure formulas of chemical substances in this paper needs to be further corrected and standardized.

Answer: We thank the reviewer for the suggestion; we have modified.

  1. The use of units in this paper is not standardized, for example, ml and mL are mixed, and there is a lack of uniformity in the presence or absence of spaces in front of the units, etc., which needs to be further revised.

Answer: We thank the reviewer; we have revised accordingly.

  1. It is recommended that the authors reorder Tables 2 and 3 in this paper; the order of the tables should be in the order in which they appear in the text.

Answer: We have reordered the tables as suggested

  1. It is recommended that the authors add the names of the corresponding fatty acids in Table 3, which would be more clear and concise, rather than just providing them in the table notes.

Answer: We thank the reviewer for the suggestion; we have modified.

  1. The presentation of Table 5 is not standardized and the authors are advised to revise it to a 3-line table.

Answer: We have harmonized the table.

  1. After "3. Results and Discussion" is "5. Conclusions"? It is suggested that the authors amend the expression.

Answer:  We have corrected it.

  1. The standardized use of abbreviations, units, and chemical structure formulas, etc. in this paper needs to be further corrected.

Answer: Done.